# Women in academic surgery over the last four decades

**Laura J. Linscheid**[1⊕], **Emma B. Holliday**[2⊕], **Awad Ahmed**[3], **Jeremy S. Somerson**[1], **Summer Hanson**[4], **Reshma Jagsi**[5], **Curtiland Deville, Jr**[6]*

**1** Department of Orthopaedic Surgery and Rehabilitation, The University of Texas Medical Branch, Galveston, Texas, United States of America, **2** Department of Radiation Oncology, The University of Texas MD Anderson Cancer Center, Houston, Texas, United States of America, **3** Department of Radiation Oncology, MercyOne, Waterloo, Iowa, United States of America, **4** Department of Plastic Surgery, The University of Texas MD Anderson Cancer Center, Houston, Texas, United States of America, **5** Department of Radiation Oncology, The University of Michigan, Ann Arbor, Michigan, United States of America, **6** Department of Radiation Oncology, Johns Hopkins Kimmel Cancer Center, Baltimore, Maryland, United States of America

⊕ These authors contributed equally to this work.

* cdeville@jhmi.edu

**Data Availability Statement:** The data underlying the results presented in this study are available from The Association of American Medical Colleges (AAMC) at https://www.aamc.org/what-

## Abstract

### Objective

As the number of female medical students and surgical residents increases, the increasing number of female academic surgeons has been disproportionate. The purpose of this brief report is to evaluate the AAMC data from 1969 to 2018 to compare the level of female academic faculty representation for surgical specialties over the past four decades.

### Design

The number of women as a percentage of the total surgeons per year were recorded for each year from 1969–2018, the most recent year available. Descriptive statistics were performed. Poisson regression examined the percentage of women in each field as the outcome of interest with the year and specialty (using general surgery as a reference) as two predictor variables.

### Setting

Data from the American Association of Medical Colleges (AAMC).

### Participants

All full-time academic faculty physicians in the specialties of obstetrics and gynecology (OB/GYN), general surgery, ophthalmology, otolaryngology (ENT), plastic surgery, plastic surgery, urology, neurosurgery, orthopaedic surgery and cardiothoracic surgery as per AAMC records.

we-do/mission-areas/patient-care/workforce-
studies/reports.

**Funding:** The author(s) received no specific
funding for this work.

**Competing interests:** The authors have declared
that no competing interests exist.

## Results

The percentage of women in surgery for all specialties evaluated increased from 1969 to 2018 (OR 1.04, p<0.001). Compared with general surgery, the rate of yearly percentage change increased more slowly in neurosurgery (OR 0.84; P = .004), orthopaedic surgery (OR 0.82; P = .002), urology (OR 0.59; P < .001), and cardiothoracic surgery (OR 0.38; P < .001). There was no significant difference in the rate of yearly percentage change for plastic surgery (OR 1.01; P = .840). The rate of yearly percentage change increased more rapidly in OB/GYN (OR 2.86; P < .001), ophthalmology (OR 1.79; P < .001) and ENT (OR 1.70; P < .001).

## Conclusions

Representation of women in academic surgery is increasing overall but is increasing more slowly in orthopaedic surgery, neurosurgery, cardiothoracic surgery and urology compared with that in general surgery. These data may be used to inform and further the discussion of how mentorship and sponsorship of female students and trainees interested in surgical careers may improve gender equity in the future.

## Introduction

When the "New Yorker Cover Challenge" and the #ILookLikeASurgeon hashtag went viral on social media, much attention was directed to women in surgical fields. These movements succeeded in raising awareness of gender stereotyping and highlight the disparity in representation of women in surgery. According to 2017 data from the American Association of Medical Colleges (AAMC), more than one-third of the active physician work force in the US and more than half of incoming medical school students are female; however, women still make up less than one-quarter of the faculty of 10 surgical specialties including only 5.3% of orthopedic surgeons with the lowest representation [1]. As the number of female medical students and surgical residents increases, the increasing number of female academic surgeons in certain surgical specialties has been disproportionately low.

The purpose of this brief report is to evaluate the AAMC data from 1969 to 2018 to do the following: 1.) report current numbers and percentages of women in obstetrics and gynecology (OB/GYN) general surgery, ophthalmology, otolaryngology (ENT), plastic surgery, urology, neurosurgery, orthopaedic surgery and cardiothoracic surgery; 2.) compare current level of female representation with data from the past four decades; 3.) identify rates of change for female representation in surgical subspecialties compared with general surgery as a reference.

## Methods

Institutional review board exemption was granted as primary data were obtained via publically available sources. All full-time academic faculty physicians in the specialties of OB/GYN, general surgery, ophthalmology, ENT, plastic surgery, plastic surgery, urology, neurosurgery, orthopaedic surgery and cardiothoracic surgery were obtained from the AAMC. The number of female surgeons as a percentage of the total number per year were recorded for each year between 1969 and 2018- the most recent year available. Descriptive statistics were performed. Poisson regression examined the percentage of women in each field as the outcome of interest

with the standardized year and specialty (using general surgery as a reference) as two predictor variables. Statistical analyses were performed with the statistical computing software R (R version 3.6.2—"Dark and Stormy Night", R Core Team 2018) [2].

## Results

As reported by the AAMC, in 2018, the number (percentage) of women academic faculty surgeons were 4050 (63.3%) for OB/GYN, 251 (30.8%) for general surgery, 1220 (39.8%) for ophthalmology, 825 (34.9%) for otolaryngology, 119 (26.3%) for plastic surgery, 309 (21.0%) for urology, 425 (20.8%) for neurosurgery, 790 (19.2%) for orthopaedic surgery and 115 (16.1%) for cardiothoracic surgery. Fig 1 shows the change in the percentage of women in each surgical field from 1980 to 2018.

The percentage of women in all included surgical fields increased over the time period of interest (OR 1.04, p<0.001).

With general surgery as a reference, the rate of yearly percentage change increased significantly faster for women in OB/GYN (OR 2.86; P<0.001), ophthalmology (OR 1.7; P < .001) and ENT (OR 1.7; P < .001). The rate of yearly percentage change was not significantly different for plastic surgery (OR 1.01; P = .840). However, the rate of yearly percentage change increased significantly slower for women in neurosurgery (OR 0.84; P = .004), orthopaedic surgery (OR 0.82; P = .002), urology (OR 0.59; P < .001) and cardiothoracic surgery (OR 0.38; P < .001).

## Discussion

The percentage of women in medical school has increased greatly since 1970 when women made up less than 6 percent of medical student populations [3]. According to the AAMC, women have made up the majority of incoming medical students for the past two years with 50.7% in 2017 and 51.6% in 2018 of incoming medical students being female. Female representation in surgical specialties has increased over this time period as well but the percentage of female surgeons and surgery residents still hovers around 30% and women make up less than 20% of surgical leadership positions [4]. As a whole, the number of women in academic surgery is increasing, but certain surgical specialties have shown greater increases in the representation of women compared to others.

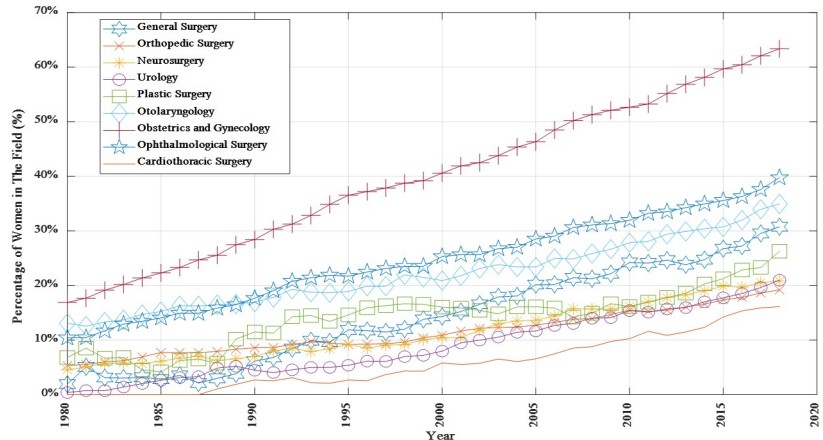

**Fig 1. Percentage of women among practicing academic surgeons from 1980 to 2018.**

There are several common barriers to women choosing to pursue academic surgical careers that impact all surgical specialties. These include a lack of female mentors in surgery, perceived poor work-life balance, gender stereotyping, and harassment. Positive role models with similar interests and goals can be a crucial factor in medical students' career choices. For example, female first authors are more likely then male first authors to publish with female senior authors [5]. Women make up less than 20% of full time surgical faculty and only 7.7% of surgery chairs so often female medical students may not easily have access to same-sex mentors [6]. Another deterrent to women choosing a surgical field may be the perceptions of a surgical lifestyle and its effects on life outside the hospital or delay their plans for a family. Female surgeons are more likely than their male counterparts to still be responsible for household tasks such as childcare planning, meal planning, and grocery shopping [7]. Female surgical residents also were more likely to not have children or to have deliberately delayed childbearing until after medical school or residency [7]. Additionally, female surgical trainees report gender bias or discrimination still affects their medical school experience or surgical training. Women in male-dominated specialties were significantly more likely to report that, due to gender discrimination, they would not recommend their field to a trainee or family member and reported being more likely to leave medicine or retire early [8].

The differences in the trajectories of women entering the various academic surgical specialties studied is interesting and merits further study. OB/GYN is the clear outlier in our study sample with a rate of increase nearly three times higher than that of general surgery. Women currently make up the majority of OB/GYNs, and there has been much discussion about this trend and its implications on the pay and reputation of the field. Labor economics research has shown a "tipping point"; when 30% of positions in a given field are occupied by women, it starts to be seen as "women's work" and men start leaving the field at even faster rates [9]. Based on our dataset, the field of OB/GYN reached that "tipping point" of 30% female representation in 1991. The percentage of women within the field of OB/GYN is projected to continue its upward trajectory. A recent evaluation of residents showed women make up 82.5% of OB/GYN trainees [10]. It is possible that a combination of factors including decreased interest in OB/GYN from male medical students and an increased number of available mentors and role models for female medical students accelerated the proportion of female surgeons in this field. To a lesser extent, women are also entering the fields of ophthalmology and ENT at a rate faster than general surgery. Workforce and supply/demand factors may be at least partially driving the increasing representation of women in ophthalmology [11]. Additionally, women in ophthalmology may have increased numbers of visible role models and available mentors as studies have shown the percentage of women on journal editoral boards and professional society boards are at least in line with the percentage of women in the field as a whole [12]. There is similar gender parity present in the leadership positions of ENT societies relative to the percentage of practicing female ENT surgeons [13]. The development of robust and active mentorship programs for students and residents may be helping to relieve pipeline issues [14].

The rate of change for women in academic plastic surgery is also increasing relatively relative to many other specialties studied. This may be, in part, driven by the fact that women make up the majority of patients receiving plastic surgery services and express an increasing preference for a female surgeon [15]. Finally, OB/GUN, ophthalmology, ENT and plastic surgery all have increased opportunities for a majority outpatient or elective surgical practice. There may also be increased opportunities for flexibility with evening and weekend work responsibilities which may be appealing to women interested in surgery who are also faced with caretaking or other domestic responsibilities. General surgery was our reference group for this study, mainly because its rate of change over the time period studied was in the middle of the overall cohort. This may be due to the heterogeneity within general surgery practices. Subspecialties within

general surgery have their own cultures, call and inpatient coverage expectations as well as availability of female mentors and sponsors.

On the other end of the spectrum, the surgical specialties of neurosurgery, orthopaedic surgery, urology and cardiothoracic surgery saw the representation of women within their academic workforce grow more slowly over the time period studied. The four specialties also have low rates of women in leadership positions at the chair and national professional society levels [16]. Interestingly, tradition and departmental culture may also propagate gender stereotypes that women do not "belong" in these surgical specialties. A study of gendered language on departmental websites showed orthopaedics and neurosurgery departments were more likely to use gendered language (such as "chairman") compared to more gender neutral language (such as "chair") [17]. Similar to what is seen in other specialties, female surgeons take on more childcare and domestic responsibilities at home [18]. This may make surgical specialties with increased call responsibility, inpatient coverage, and unpredictable schedules less desirable to women entering surgery.

Several surgical specialties are proactively developing programs to aid in the recruitment and retention of female surgeons. Often cited action items include: identifying and eliminating discrimination in recruitment and hiring of residents and faculty surgeons, fostering and mentoring women in their career advancement, and promoting qualified women to visible leadership positions [19]. There are several other strategies that have been studied as well. Unconscious bias training has been shown to be an effective tool in combating gender bias and promoting gender equity, and should be considered as an addition to academic centers training curriculum [20]. Large numbers of academic centers lack formal programs for the recruitment, promotion, or retention of women in academic medicine. Policies and standards set by institutions such as the AAMC regarding development of such formal programs could promote a culture that enhances female recruitment and encourages a strong network of women faculty [21]. Increasing female surgical faculty and women in leadership positions would provide medical students with more access to positive female role models in these male dominated areas and could lead to more women choosing to go into surgical fields. It is important to acknowledge that the factors involved in specialty choice are multiple, complicated and personal for each medical student. Successfully implementing programs targeted at the potentially modifiable factors mentioned above is unlikely to result in 50% representation of women in all surgical specialties, but removing barriers that capable, interested and motivated women disproportionately face to achieving a successful surgical career would be, in and of itself, a lofty and worthwhile goal.

Strengths of this study include the large sample size of surgeons we were able to obtain by accessing AAMC data. However, our interpretation is limited by inherent limitations faculty classification within the AAMC database. Specifically, the number of general surgery faculty is likely artificially low in this database due to discrepancies departmental affiliation as reported to the AAMC.

In conclusion, representation of women in surgical specialties is increasing more slowly in neurosurgery, orthopaedic surgery, urology and cardiothoracic surgery than in other specialties. This detailed and focused analysis of time trends and patterns of women's representation in surgical specialties highlights the ongoing need for evidence-based interventions to promote gender equity. These data provide a benchmark moving forward to allow departmental and institutional leadership to measure gender diversity within their own faculty against the national average. They may also be used to inform and further the discussion of how mentorship and sponsorship of female students and trainees interested in surgical careers may improve gender equity in the future.

## Supporting information

**S1 Data. Data from the Association of American Medical Colleges outlining the number of men and women practicing in the academic surgical specialties of obstetrics and gynecology, general surgery, ophthalmology, otolaryngology, plastic surgery, plastic surgery, urology, neurosurgery, orthopaedic surgery and cardiothoracic surgery from 1969 to 2018.** (CSV)

## Author Contributions

**Conceptualization:** Awad Ahmed, Summer Hanson, Reshma Jagsi, Curtiland Deville, Jr.

**Data curation:** Awad Ahmed, Jeremy S. Somerson, Curtiland Deville, Jr.

**Formal analysis:** Awad Ahmed.

**Methodology:** Awad Ahmed, Curtiland Deville, Jr.

**Project administration:** Awad Ahmed, Curtiland Deville, Jr.

**Supervision:** Emma B. Holliday, Jeremy S. Somerson, Curtiland Deville, Jr.

**Writing – original draft:** Laura J. Linscheid, Emma B. Holliday.

**Writing – review & editing:** Laura J. Linscheid, Emma B. Holliday, Awad Ahmed, Jeremy S. Somerson, Summer Hanson, Reshma Jagsi, Curtiland Deville, Jr.

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
