## [Decision Letter · Decision Letter 0]

6 Nov 2020

PONE-D-20-32410

Women in Academic Surgery Over the Last Four Decades Unequal Rate of Change Among Surgical Specialties

PLOS ONE

Dear Dr. Holliday,

Thank you for submitting your manuscript to PLOS ONE. After careful consideration, we feel that it has merit but does not fully meet PLOS ONE’s publication criteria as it currently stands. Therefore, we invite you to submit a revised version of the manuscript that addresses the points raised during the review process.

Please address comments from the review.

We look forward to receiving your revised manuscript.

Kind regards,

Leonidas G Koniaris, MD

Academic Editor

PLOS ONE

Journal Requirements:

2. Please list the name and version of any software package used for statistical analysis, alongside any relevant references. For more information on PLOS ONE's expectations for statistical reporting, please see https://journals.plos.org/plosone/s/submission-guidelines.#loc-statistical-reporting.

Reviewers' comments:

Reviewer's Responses to Questions

**Comments to the Author**

1. Is the manuscript technically sound, and do the data support the conclusions?

Reviewer #1: Yes

2. Has the statistical analysis been performed appropriately and rigorously? 

Reviewer #1: Yes

3. Have the authors made all data underlying the findings in their manuscript fully available?

Reviewer #1: Yes

4. Is the manuscript presented in an intelligible fashion and written in standard English?

Reviewer #1: Yes

5. Review Comments to the Author

Reviewer #1: In the article “Women in academic surgery over the last four decades unequal rate of change among surgical specialties” the authors review the AAMC data on women faculty in various surgical specialties. As the proportion of female medical school matriculants approach (and slightly surpass) gender equity, the rate of change of women faculty in academic surgery is evaluated. The study design is simple and its execution is clean. This manuscript does provide a good benchmark from which to work toward gender equality.

Major

1. The authors did not include the discipline of gynecology. While this field is more female predominant than other surgical specialties, the proportion of female faculty (and trends there in) would regardless be interesting. Would the authors comment on their rationale for not including gynecology in this study.

2. As the authors suggest, role modeling and mentorship often play critical roles in recruitment. In order to accomplish this, there needs to be a critical mass of mentors/role models. Once there is a critical mass, the rate of change/growth can become synergistic and exponential. Perhaps the rate of change in ophthalmology and otolaryngology is greater is because at the start of the study period that “critical mass” (10%?) was already achieved. Cardiothoracic surgery, neurosurgery, and urologic surgery had close to 0% female faculty in 1980 and only reach 10% around year 2000-2010. Perhaps at this point, the rate of change will start to catch up with the other surgical specialties (and that of opthalmology & OHNS will start to level out). Do the authors note an “inflection point” or value at which this “critical mass” is reached?

3. While it is nice to have a goal of gender equality, perhaps gender equity is the better objective. Despite female medical school matriculants representing 50% of medical school classes, perhaps female medical students do not care as much to be in cardiothoracic surgery, urology or neurosurgery and no amount of mentorship/role modeling/etc will ever achieve a work force in these fields of 50% women. Perhaps we as a community should be at peace with this concept and should cease to feed women the pressure that they “can have it all” and “should have it all” and be in demanding fields (such as cardiothoracic surgery, neurosurgery) in addition to all the additional burdens of life outside of work (such as running a household, etc). While women should have equal opportunities to enter these fields (equality) they should have the freedom to choose a career path that is right for their life balance (equity). Would the authors please comment on the fact that perhaps certain surgical specialties are simply not attractive to medical students as a career path, and that these choices may be independent of gender biases within the specialties?

Minor

1. Methods/3rd line—plastic surgery is repeated twice.

2. Discussion/paragraph 3—The sentence “The rate of change for women in plastic surgery is also increasing relatively relative to many other specialties studied” does not read well. Recommend editing that statement (eg, “The rate of change for women in plastic surgery is also increasing relative to many other specialties.”)

6. PLOS authors have the option to publish the peer review history of their article (what does this mean?). If published, this will include your full peer review and any attached files.

Reviewer #1: No

---

## [Author Response · Author response to Decision Letter 0]

17 Nov 2020

Reviewer Comments to the Author  Reviewer #1: In the article “Women in academic surgery over the last four decades unequal rate of change among surgical specialties” the authors review the AAMC data on women faculty in various surgical specialties. As the proportion of female medical school matriculants approach (and slightly surpass) gender equity, the rate of change of women faculty in academic surgery is evaluated. The study design is simple and its execution is clean. This manuscript does provide a good benchmark from which to work toward gender equality. 

Major 1. The authors did not include the discipline of gynecology. While this field is more female predominant than other surgical specialties, the proportion of female faculty (and trends there in) would regardless be interesting. Would the authors comment on their rationale for not including gynecology in this study.

-The reviewer raises an excellent point. We have included the discipline of Obstetrics and Gynecology in our revised analysis. 

 

2. As the authors suggest, role modeling and mentorship often play critical roles in recruitment. In order to accomplish this, there needs to be a critical mass of mentors/role models. Once there is a critical mass, the rate of change/growth can become synergistic and exponential. Perhaps the rate of change in ophthalmology and otolaryngology is greater is because at the start of the study period that “critical mass” (10%?) was already achieved. Cardiothoracic surgery, neurosurgery, and urologic surgery had close to 0% female faculty in 1980 and only reach 10% around year 2000-2010. Perhaps at this point, the rate of change will start to catch up with the other surgical specialties (and that of opthalmology & OHNS will start to level out). Do the authors note an “inflection point” or value at which this “critical mass” is reached?

-The reviewer raises an excellent point, and we agree there probably is a critical mass of women with in a field necessary to be sufficiently visible and available for mentorship, sponsorship and role modeling. Although we were not able to identify such an “inflection point’ in our data, there are older labor economics studies that show once women comprise approximately 30% of the workforce in a given profession, men tend to increasing leave the field as it gains a reputation as “women’s work” leading to decrease prestige and pay. We highlighted the sociology data here in the discussion. 

 

3. While it is nice to have a goal of gender equality, perhaps gender equity is the better objective. Despite female medical school matriculants representing 50% of medical school classes, perhaps female medical students do not care as much to be in cardiothoracic surgery, urology or neurosurgery and no amount of mentorship/role modeling/etc will ever achieve a work force in these fields of 50% women. Perhaps we as a community should be at peace with this concept and should cease to feed women the pressure that they “can have it all” and “should have it all” and be in demanding fields (such as cardiothoracic surgery, neurosurgery) in addition to all the additional burdens of life outside of work (such as running a household, etc). While women should have equal opportunities to enter these fields (equality) they should have the freedom to choose a career path that is right for their life balance (equity). Would the authors please comment on the fact that perhaps certain surgical specialties are simply not attractive to medical students as a career path, and that these choices may be independent of gender biases within the specialties?

-We completely agree that gender equity, and not gender equality should be the goal. All medical students with interest in and aptitude for a surgical career should have access to opportunities, mentors and support to achieve their goals. Potentially modifiable factors such as lack of mentorship and toxic workplace cultures that foster gender discrimination and harassment should be the focus with discrete interventions such as mentorship programs and culture change. The reviewer brings up an excellent point that there are many reasons why a young woman in her 20s or 30s may not want to choose a surgical specialty given the current culture and work-life balance. We included in our discussion that women are more likely to have partners who work full time outside the home and are still more likely to be responsible for the majority of domestic and caretaking responsibilities inside the home. Until there is a much more broad societal change with regard to traditional gender roles, the preferences and priorities of male and female medical students are likely never going to be “equal”.  

Minor 1. Methods/3rd line—plastic surgery is repeated twice.

-Thank you, this has been corrected. 

 2. Discussion/paragraph 3—The sentence “The rate of change for women in plastic surgery is also increasing relatively relative to many other specialties studied” does not read well. Recommend editing that statement (eg, “The rate of change for women in plastic surgery is also increasing relative to many other specialties.”)

-Thank you, this has been corrected.

---

## [Editor Report · Decision Letter 1]

19 Nov 2020

Women in Academic Surgery Over the Last Four Decades

PONE-D-20-32410R1

Dear Dr. Holliday,

We’re pleased to inform you that your manuscript has been judged scientifically suitable for publication and will be formally accepted for publication once it meets all outstanding technical requirements.

Kind regards,

Leonidas G Koniaris, MD

Academic Editor

PLOS ONE
---

## [Editor Report · Acceptance letter]

23 Nov 2020

PONE-D-20-32410R1 

Women in Academic Surgery Over the Last Four Decades: 

Dear Dr. Holliday:

I'm pleased to inform you that your manuscript has been deemed suitable for publication in PLOS ONE. Congratulations! Your manuscript is now with our production department. 

Kind regards, 

on behalf of

Dr. Leonidas G Koniaris 

Academic Editor

PLOS ONE